# The impact of radiofrequency exposure on *Aedes aegypti* (Diptera: Culicidae) development

**Nik Muhammad Hanif Nik Abdull Halim[1,2], Alya Farzana Mohd Jamili[1], Nazri Che Dom[1,3,4], Nurul Huda Abd Rahman[5], Zana Jamal Kareem[6,7], Rahmat Dapari**[8] *

**1** Centre of Environmental Health & Safety, Faculty of Health Sciences, Universiti Teknologi MARA (UiTM), UITM Cawangan Selangor, Puncak Alam, Selangor, Malaysia, **2** Integrated Mosquito Research Group (I-MeRGe), Universiti Teknologi MARA (UiTM), UITM Cawangan Selangor, Puncak Alam, Selangor, Malaysia, **3** Institute for Biodiversity and Sustainable Development (IBSD), Universiti Teknologi MARA, Shah Alam, Malaysia, **4** Setiu District Health Office, Permaisuri, Terengganu, Malaysia, **5** Antenna Research Centre, School of Electrical Engineering, College of Engineering, Universiti Teknologi MARA, Shah Alam, Selangor, Malaysia, **6** Faculty of Health Sciences, Qaiwan International University, Sulaymaniyah, Iraq, **7** Kurdistan Institution for Strategic Studies and Scientific Research (KISSR), Sulaymaniyah, Iraq, **8** Department of Community Health, Universiti Putra Malaysia, Serdang, Selangor, Malaysia

* drrahmat@upm.edu.my

## Abstract

### Introduction

Wireless communication connects billions of people worldwide, relying on radiofrequency electromagnetic fields (RF-EMF). Generally, fifth-generation (5G) networks shift RF carriers to higher frequencies. Although radio, cell phones, and television have benefitted humans for decades, higher carrier frequencies can present potential health risks. Insects closely associated with humans (such as mosquitoes) can undergo increased RF absorption and dielectric heating. This process inadvertently impacts the insects' behaviour, morphology, and physiology, which can influence their spread. Therefore, this study examined the impact of RF exposure on *Ae. aegypti* mosquitoes, which are prevalent in indoor environments with higher RF exposure risk. The morphologies of *Ae. aegypti* eggs and their developments into *Ae. aegypti* mosquitoes were investigated.

### Methods

A total of 30 eggs were exposed to RF radiation at three frequencies: baseline, 900 MHz, and 18 GHz. Each frequency was tested in triplicate. Several parameters were assessed through daily observations in an insectarium, including hatching responses, development times, larval numbers, and pupation periods until the emergence of adult insects.

### Results

This study revealed that the hatching rate for the 900 MHz group was the highest (79 ± 10.54%) compared to other exposures ($p = 0.87$). The adult emergence rate for the 900 MHz group was also the lowest at 33 ± 2.77%. A significant difference between the groups was demonstrated in the statistical analysis ($p = 0.03$).

**Data Availability Statement:** All relevant data are within the manuscript.

**Funding:** The author(s) received no specific funding for this work.

**Competing interests:** The authors have declared that no competing interests exist.

## Conclusion

This work highlighted the morphology sensitivity of *Ae. aegypti* eggs and their developments in the aquatic phase to RF radiation, potentially altering their life cycle.

## Introduction

Mosquitoes are of great public health significance as significant disease vectors that affect human populations worldwide, particularly the *Aedes* species. *Aedes* mosquitoes are diurnal insects thriving in tropical and subtropical locations, which are responsible for transmitting dengue fever [1]. These mosquitoes possess a preference for habitats that are near humans and are highly efficient in transferring arboviruses. The *Aedes* (Stegomyia) *aegypti* (Linnaeus) (Diptera: Culicidae) is a prominent dengue vector primarily present in urban settings globally. Various environmental factors significantly affect the development rate of *Aedes* mosquitoes, including biotic and abiotic components [2]. Hence, a combination of climate, geography, environment, and social conditions can produce an optimal environment for the survival of the vector species and the dengue virus.

The *Ae. aegypti* mosquito (yellow fever mosquito) is pivotal in transmitting dengue fever, while the *Ae. albopictus* mosquito (Asian tiger mosquito) is a secondary vector for this disease [3]. Given that mosquitoes possess a complex life cycle intricately connected to human environments, they are anthropophilic. These mosquitoes strongly associate with human habitats and can quickly enter buildings for feeding and resting. The female *Ae. aegypti* mosquitoes lay their eggs in containers filled with clear water and a minimal organic presence, which are positioned on moist surfaces just above the water level. Each female mosquito can deposit approximately 100 eggs, engaging in this reproductive cycle three times throughout her lifespan [4]. These oval-shaped eggs measure 1 mm long and are slightly larger than the *Ae. albopictus* eggs.

An in-depth comprehension of mosquito life cycles is essential for gaining insights into population biology, dynamics, vector capacity, and their impact on the disease outbreak risk. Hence, traditional methods have been employed to manage *Aedes* mosquito populations efficiently, including source reduction, larvicides, and biological control. Innovative approaches include using *Wolbachia* bacteria to impede virus transmission through genetic control, applying spatial repellents, and implementing GIS with remote control for larval source management. These strategies are aimed at achieving a comprehensive approach to mosquito control.

Wireless communication relies on radiofrequency electromagnetic fields (RF-EMF) to connect billions of people globally. Therefore, introducing fifth-generation (5G) networks demonstrates a partial shift of carrier frequencies to higher ranges. Existing telecommunications networks usually operate within the frequency range of 0.1 to 6 GHz [5]. Nevertheless, 5G networks operate at frequencies up to 300 GHz. These higher frequencies are close to the millimetre wave frequency range, in which the radiation wavelength reaches a magnitude similar to the size of an insect's body [6]. Even though the impact of RF radiation on wildlife remains unknown, current studies have been effectively exploring its potential effects on various living organisms. Numerous studies have implied that environmental exposure to RF magnetic fields can impact the development and survival of certain species. This observation is particularly true in urban areas and near base stations [7].

Several significant experimental findings have successfully presented the RF exposure effects on insects. A study by Panagopoulos *et al.* reported that fruit flies exhibited reproductive capacity when exposed to 900 MHz RF radiation [8]. Likewise, a study by Vácha *et al.*

denoted a weakening of bee colonies and a disruption in their capacity to return to their hives [9]. Meanwhile, Sahib *et al.* documented impaired navigation and migration patterns in cockroaches [10]. The same author also proposed that radiation emitted by telecommunication antennae hindered honeybees' ability to navigate. This observation was attributed to the interference with the magnetite structures in the bees' bodies, which functioned as a built-in compass [10].

Another study by Cammaerts *et al.* correlated RF exposure and smell-sensing disruptions in ants [11]. Other studies also emphasised the impact of RF exposure on microbial communities [12]. A study by Poh *et al.* involved the collective positioning behaviour concerning *Ae. aegypti* mosquitoes under relatively low-power RF exposure. The study yielded inconclusive results regarding its impact on flight ability and direction [13]. Alternatively, a study by Borre *et al.* examined higher-frequency RF exposures in adult *Ae. aegypti* mosquitoes. The study revealed that the exposure developed dielectric heating, which affected the physiology, behaviour, and morphology of adult *Ae. aegypti* mosquitoes [14]. Multiple scientific studies have also highlighted the detrimental impact of RF exposure on various insect taxa. Nonetheless, insufficient knowledge about the consequences of anthropogenic RF radiation on indoor-adapted mosquitoes (such as *Ae. aegypti* mosquitoes) has remained limited.

The influence of RF exposure on the development of *Aedes* mosquito populations (particularly *Ae. aegypti* mosquitoes) has not been effectively explored. These factors can influence the *Aedes* population's growth and affect dengue transmission severity. Thus, this study investigated the impact of RF exposure on the life cycle of *Ae. aegypti* mosquitoes. Considering that the *Aedes* species underwent holometabolous development involving distinct stages (egg to larva, pupa, and adult), the RF radiation influence on the developmental characteristics of *Ae. aegypti* mosquitoes from the early stages to adult mosquitoes were assessed.

## Methods

### Mosquito strain collection

The laboratory strains of *Ae. aegypti* eggs were obtained from the Department of Medical Entomology, Infectious Diseases Research Centre, Institute of Medical Research, Kuala Lumpur, Malaysia. An ovitrap surveillance equipment collected field strain eggs over 16 weeks from March 2022 to June 2022. This equipment was installed in places with established *Aedes* populations in Puncak Alam, Selangor, Malaysia [15, 16]. The *Ae. aegypti* egg collection process adhered to the protocols outlined in Faull *et al.*'s study [17]. Initially, the ovitraps were constructed using black plastic containers with a 4 cm-wide central opening (8 cm in diameter and 11 cm in height). These ovitraps were reinforced with a 14 cm × 3 cm paddle and utilised as oviposition sites for *Aedes* females. Approximately two-thirds of the ovitraps were filled with water.

### Colonisation and mosquito cohort establishment

The mosquitoes from natural environments and the laboratory were raised and maintained in the vector insectary at the Faculty of Health Sciences, UiTM Puncak Alam. This insectary was carefully controlled with a temperature of 29 ± 3˚C, 75 ± 10% relative humidity (RH), and a 12:12 h light-dark cycle. The field strain eggs laid on the ovitraps paddle were submerged in containers containing deionised water, facilitating their hatching. Approximately four to seven days after hatching, three to five larvae were randomly chosen and collected for identification using a pipette. These larvae were briefly euthanised in hot water, placed on a glass slide, covered with a coverslip, and examined under a stereomicroscope to achieve precise taxonomic identification.

The *Aedes* species were identified during the larval stage, and caution was exercised to prevent disfigurement while handling the deceased larvae. Several key taxonomic features were applied for the identification, including a short with a stout syphon, a row of thorn-like crest scales on abdominal segment VII, and the absence of hooks on the side of the thorax. These features were based on two studies by Rueda and Christopher [18, 19]. The *Ae. aegypti* eggs were immersed in deionised water supplemented with nutrients, and the larval growth was monitored daily. Fine fish food (TetraMin Plus Tropical Fish Flakes) was also provided to facilitate the development of the recently hatched larvae. This larvae were fed daily throughout all phases until they reached the pupation stage. The food given to each larva increased progressively daily, consisting of 0.06 to 0.12 mg per larva in L1 to L2, 0.24 mg in L3, and 0.48 mg in L4.

Daily water changes were conducted to maintain optimal conditions and prevent tank foaming and volume loss. Sex separation was then performed at the pupal stage to obtain virgin mosquitoes, taking advantage of the size difference between the female and male pupae. Approximately 50 pupae were segregated into batches and placed in plastic containers covered with fine mesh. Each container contained 10 ml of dechlorinated water. A total of 15 pairs of virgin adult mosquitoes were introduced in standard rearing cages (30 × 30 × 30 cm), in which a solution containing 10% sucrose was administered as a nutrient source. After a 72-h mating period, a blood meal was administered by introducing a confined white mouse into the mosquito cage for 12 h. This process was performed to ensure that the female mosquitoes were sexually mature and had laid eggs.

An egg-laying site was established within the cage following the blood meal. This process was achieved by placing plastic containers (9 cm in height and 7.5 cm in diameter) filled with 100 ml of water and filter paper as the substrate. Subsequently, the strips containing eggs were gathered and substituted daily for one week. This cycle of larval development, adult mating, and oviposition was repeated until enough $F_2$ generation was obtained for the laboratory and field strains. Moreover, this study exclusively utilised $F_2$ generation mosquitoes to determine whether the desirable traits remained consistent in consecutive generations or displayed unexpected changes [20]. The eggs were left to undergo embryonation for two days after collection. These eggs were then dried on paper towels for 30 s and stored in a moist environment for an extra day.

Rather than being fully dry, the eggs were dried at rearing temperatures until they reached a slightly damp state. These eggs were then stored in airtight bags with moist absorbent cotton to maintain high humidity and prevent desiccation. This study utilised the $F_2$ eggs after a storage time between two and three weeks. Typically, the age of egg batches potentially influences the hatching response. Given that the eggs presented inconsistent age, the age factor was considered an exclusion criterion. This study required 1620 eggs from the laboratory and field strains to conduct a single triplicate experiment to investigate the effect of RF exposure on egg hatching and larval development in *Ae. aegypti* mosquitoes.

## Configuration of the exposure chamber and antenna

Fig 1 depicts the experimental chamber comprising a large polystyrene foam box (49.5 cm × 62 cm × 37 cm) and a thickness of 3 cm. The entire chamber was uniformly coated in white paint to enhance the contrast of the samples. This configuration was composed of seven key compartments: (A) power supply, (B) analogue signal generator, (C) spectrum analyser, (D) RF antenna, (E) exposure chamber, (F) sample, and (G) RF cable. A wideband antenna (A-Info, LB8180-NF model, J203020295; Irvine, CA, USA) with frequency-dependent radiation characteristics was employed in this configuration. This antenna featured a frequency

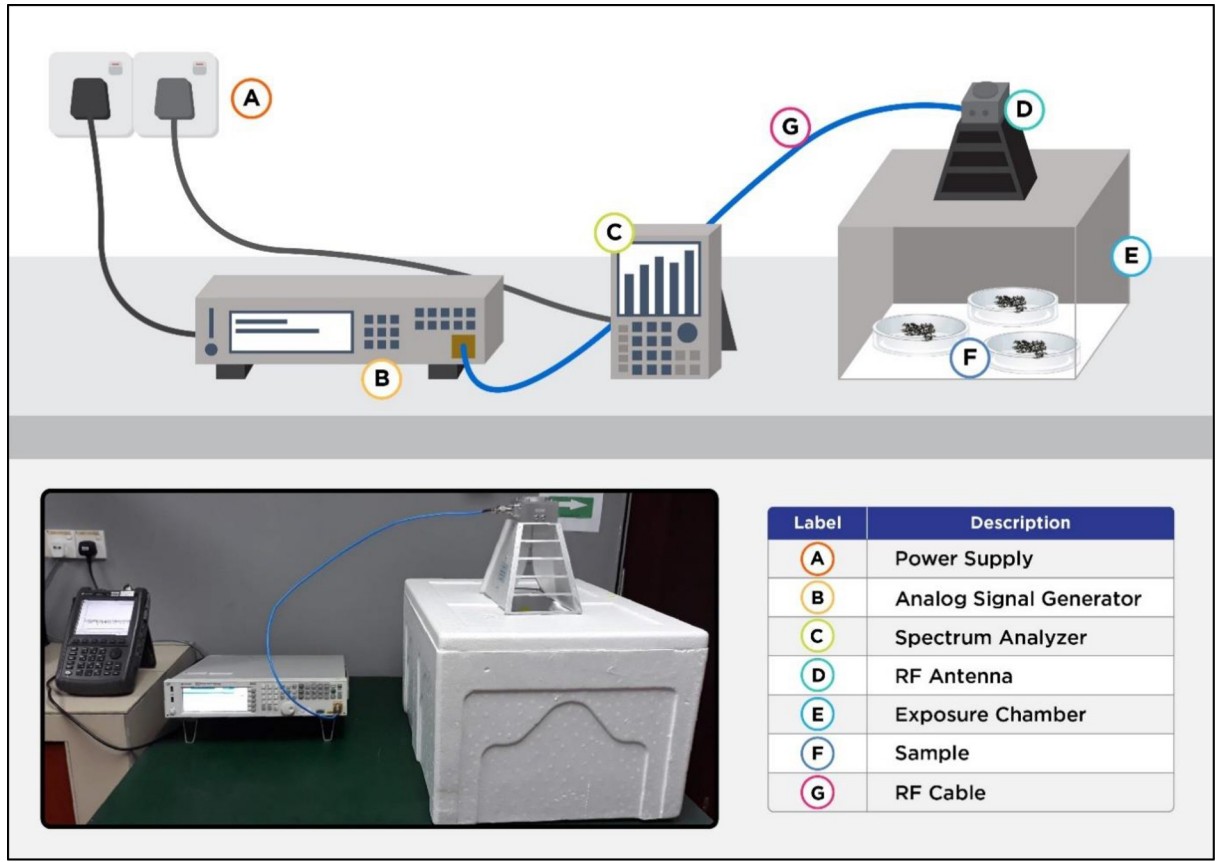

**Fig 1. The designated chamber and antenna for RF exposure on *Ae. aegypti* eggs.**

range from 0.8 to 18 GHz, an average gain of 12 dB, and linear polarisation. The antenna weighed approximately 1.5 kg and was used to transmit the generated RF waves to the test subjects. Consequently, a single antenna was enough to radiate power across various frequencies due to its wide bandwidth.

The antenna was positioned within the designated compartment and oriented towards the intervention site. This study applied a wideband function generator (Keysight, model N5173B EXG X-Series; Colorado Springs, CO, USA) as the RF wave source, which could produce high-frequency signals from 9 kHz to 40 GHz. A super ultra-wideband amplifier was also connected to the signal generator using a coaxial cable with SMA connectors to enhance the signal. Finally, an amplifier was connected to the antenna using a coaxial cable, with both cables terminated with SMA-to-N connectors.

## The RF exposure protocols

The $F_2$ generation eggs of the *Ae. aegypti* strain were randomly assigned to the experimental group to eliminate any potential influence of egg age on the outcomes. Before RF exposure, the eggs were meticulously inspected for defects under a stereomicroscope (Olympus, model Courtesy; UK Ltd). Subsequently, the eggs were placed on filter paper inside plastic containers (16.5 cm × 7.5 cm × 10 cm) with lids and stored at room temperature. Approximately 90 mosquito eggs were exposed to two RF frequencies: 900 MHz and 18 GHz. Each exposure lasted for 24 h in dedicated chambers.

The RF exposure process was replicated three times (in triplicate) to establish the validity and reliability of results for the laboratory and field strains of *Ae. aegypti* mosquitoes. These replications co-occurred on the same day. Meanwhile, the procedures were duplicated without any RF exposure as the control group. A single RF exposure for the *Ae. aegypti* eggs was also necessary to assess mosquito development.

## Experimental design

A completely randomised factorial 2 × 2 × 3 design was employed in this study, resulting in a total of 12 treatments. Two exposure duration levels were observed in the independent variables: a 24-h exposure and a control group with no exposure (see Table 1). This study also incorporated *Ae. aegypti* population variations as a separate factor, which could be affected by the exposure duration. Three replications entailing the complete reproduction of the entire experiment on three different occasions were included to ensure the robustness of the findings. Each combination of these independent variables corresponded to a distinct treatment condition, leading to a diverse set of 12 treatments. A completely randomised design ensured that the treatments were assigned to experimental units in a random and unbiased manner, enhancing the reliability and validity of the results.

This study allowed for a systematic and controlled examination of exposure duration and population variation impacts on the *Ae. aegypti* population, enhancing comprehensiveness and reliability. Approximately 30 eggs from the field and laboratory strains of *Ae. aegypti* mosquitoes were used for each replicate for the egg weighing analysis. This process included three group types: unexposed (control), 900 MHz, and 18 GHz. Each of the eggs during each session was placed inside 1.5 ml Eppendorf tubes and measured using a high-precision electronic microbalance scale (Sartorius, model MPR 5; Goettingen, Germany) with an accuracy of 0.0001 g. The mean weights were computed by weighing all the egg samples to assess the sample variations after RF exposure.

An inverted microscope (Olympus, model IX73; UK Ltd) was utilised to examine the eggs for the egg morphology assessment of *Ae. aegypti* mosquitoes. After the images of the eggs were captured and transferred to a computer, the eggs were photographed using an image documentation system (DinoCapture 2.0) [21]. An Image J software was then applied to analyse the morphometric measurements [22, 23]. The software measured at least nine eggs from each RF exposure level and control. Multiple morphological parameters were evaluated, including length, width, egg index, and surface area. The egg length was determined as the distance

**Table 1. Summary of the experimental design parameters.**

| Factors | | Replication of the experiment/ number of eggs | | | Total number of eggs |
|---|---|---|---|---|---|
| Field strain | Lab strain | I | II | III | |
| (-) [a] | (-) [a] | 30 | 30 | 30 | 90 |
| (+) [b] | (-) [a] | 30 | 30 | 30 | 90 |
| (-) [a] | (+) [b] | 30 | 30 | 30 | 90 |
| (+) [b] | (+) [b] | 30 | 30 | 30 | 90 |
| **Total eggs required/strain** | | | | | **360** |

The experiment was completely randomised in a factorial 2 × 2 × 3 design (12 treatments). Independent variables comprised two exposure durations (24 h and control) on the *Ae. aegypti* population and three replications.

[a] Symbol (-) refers to the unexposed eggs (control) with RF radiation.

[b] Symbol (+) refers to the eggs exposed to RF radiation (24 h).

between the anterior and posterior poles, while the egg width was measured at the widest point [24]. Alternatively, the egg index was calculated based on the length-to-width ratio, while the surface area was automatically estimated using Image J [25].

The daily observation of the aquatic phase regarding larval development was conducted under controlled conditions of 29 ± 3˚C, 75 ± 10% RH, and a 12:12 h light-dark cycle. This study observed and measured various factors, such as the egg hatching rate, larval mortality, pupation rate, pupation duration, pupal onset ($CP_1$) and termination ($CP_2$), and adult emergence rate. The rearing temperature and conditions were also standardised at room temperature. A small amount of TetraMin Plus Tropical Fish Flakes was introduced into the tank when the eggs began to hatch. The larvae were fed daily throughout all stages until they reached the pupation stage. The feeding regime for each stage per day was 0.06 to 0.12 mg per larva in L1 to L2, 0.24 mg in L3, and 0.48 mg in L4. Additionally, the pans were inspected daily at the same hour each day. Plastic pipettes were also utilised to eliminate debris, and daily water replenishment was conducted to ensure consistent levels concerning evaporation.

Although the initial pupae generally emerge from day six to seven, they are intentionally not disturbed in the tank for more than 24 h to facilitate the larvae separation. The pupae were transferred to a labelled plastic cup (name, pupation date) during the pupation process. This cup contained clean water and was covered with a mesh using plastic pipettes. Despite the continued necessary feeding of the larvae, the food quantity was decreased upon pupation to prevent excessive contamination and foaming. Overall, the adult mosquitoes had a diet containing 10% sucrose. On the second day after the eggs were hatched, the adult *Ae. aegypti* mosquitoes were aspirated with a suction cup and placed in a labelled universal bottle. This bottle was placed in the refrigerator at 4˚C for 2 h after removing the adult mosquitoes to end their lifespans.

## Data analysis

The database of variables was analysed using IBM SPSS Statistics (version 28, copyright IBM Corporation). A parametric one-way analysis of variance (ANOVA) was employed to determine whether there were any statistically significant differences in metabolic rate measurements between the control, 900 MHz, and 18 GHz groups of the laboratory and field strains of *Ae. aegypti* mosquitoes. This study aimed to investigate the potential structural and life cycle changes in *Ae. aegypti* and *Ae. albopictus* mosquitoes caused by intense RF radiation exposure during their immature phase (eggs). The focus was examining the changes in these mosquitoes' aquatic, pupal, and adult emergence stages.

The Tukey's honestly significant difference (HSD) or Dunnet T-3 posthoc tests were performed if the results of the one-way ANOVA indicated a statistically significant difference between mosquito groups. This process determined the specific groups that differed significantly from each other. Each exposure between the laboratory and field strains was also considered concerning the comparative studies of the metabolic rate results. Therefore, the *t*-test was employed to evaluate whether there were any statistically significant disparities between the groups. This study assessed and discussed the one-way ANOVA and *t*-test study, utilising a significance value of $p < 0.05$.

## Ethics statement

This study received approval from the Committee on Animal Research and Ethics (UiTM CARE) under the reference number UiTM CARE: 377/2022 (13th May 2022). The approval duration was from July 2022 until June 2023.

**Table 2. Summary of the egg weights based on different *Ae. aegypti* strains.**

| Exposure | Strain | I | II | III | Mean | ±SD | *p*-value |
|---|---|---|---|---|---|---|---|
| **Control** | Laboratory | 13.10 | 12.11 | 11.98 | 12.40 | ±0.61 | 0.04** |
| | Field | 10.33 | 12.66 | 12.93 | 11.97 | ±1.43 | |
| **900 MHz** | Laboratory | 12.56 | 13.01 | 12.22 | 12.60 | ±0.40 | 0.15 |
| | Field | 11.25 | 11.98 | 10.89 | 11.37 | ±0.56 | |
| **18 GHz** | Laboratory | 14.01 | 10.22 | 12.45 | 12.23 | ±1.90 | 0.02** |
| | Field | 10.67 | 11.97 | 11.34 | 11.33 | ±0.65 | |

Mean weight (mg) and standard deviation (SD) for the laboratory and field *Ae. aegypti* strains concerning different doses of RF radiation levels. ** Significant difference at $p < 0.05$.

## Results

### The RF exposure effect on the weights of *Ae. aegypti* mosquito eggs

The RF radiation influence on the weight of *Ae. aegypti* eggs were provided in the tabulated data. Table 2 presents the classification of mean egg weight based on different *Ae. aegypti* strains exposed to varying RF conditions. These conditions included the control, 900 MHz, and 18 GHz exposures. Consequently, the mean egg weights of the laboratory strains demonstrated consistent and higher values than the field strain-based samples. For example, the mean egg weights of the laboratory and field strains in the control group were 12.40 ± 0.61 µg and 11.97 ± 1.43 µg, respectively. Therefore, the difference between the two strains was statistically significant ($p = 0.04$).

The mean egg weights of the laboratory and field strains in the 18 GHz RF group were 12.23 ± 1.90 µg and 11.33 ± 0.65 µg, respectively. Thus, the difference was statistically significant ($p = 0.02$). Likewise, the mean egg weights of the laboratory and field strains in the 900 MHz group were 12.60 ± 0.40 µg and 11.37 ± 0.56 µg, respectively. Conversely, the difference was not statistically significant ($p = 0.15$). When comparing the laboratory and field strains, the egg lengths between the control and the 900 MHz or 18 GHz exposure groups did not generate any statistically significant difference ($p > 0.05$). This outcome indicated that the egg weights of the laboratory and field strains remained unaffected by RF exposure.

### The RF exposure effect on the morphologies of *Ae. aegypti* mosquito eggs

Table 3 lists a comprehensive overview of the *Ae. aegypti* egg morphologies after exposure to RF radiation at 900 MHz and 18 GHz. These parameters included length (µm), width (µm), area (µm²), and egg index (µm). Consequently, significant differences between the control and

**Table 3. Summary of the egg morphology measurements based on different *Ae. aegypti* strains.**

| Exposure | Strain | Length (µm) | Width (µm) | Area (µm²) | Egg Index (µm) |
|---|---|---|---|---|---|
| **Control** | Laboratory | 745.63 ± 19.82 | 202.42 ± 16.65 | 751.80 ± 23.75 | 3.70 ± 0.24 |
| | Field | 672.71 ± 36.13 | 200.14 ± 35.26 | 703.20 ± 62.13 | 3.47 ± 0.70 |
| **900 MHz** | Laboratory | 726.87 ± 23.34 | 194.32 ± 19.48 | 726.87 ± 23.34 | 3.76 ± 0.40 |
| | Field | 702.21 ± 61.59 | 211.02 ± 52.44 | 678.35 ± 38.58 | 3.56 ± 1.09 |
| **18 GHz** | Laboratory | 726.89 ± 34.52 | 198.88 ± 27.49 | 727.92 ± 36.17 | 3.71 ± 0.50 |
| | Field | 689.49 ± 18.98 | 209.29 ± 26.21 | 685.14 ± 20.55 | 3.34 ± 0.40 |

Note: Mean egg length (µm), width (µm), area (µm²), and egg index (µm) with standard deviation (SD) for *Ae. aegypti* laboratory and field strains concerning different doses of RF radiation levels.

RF-exposed eggs from laboratory and field strains were observed. Compared to the field strain eggs, the control group from the laboratory strain exhibited more extensive length, width, area, and egg index measurements. The RF-exposed eggs from the laboratory strains at 900 MHz and 18 GHz also presented elongated lengths, larger areas, and higher egg indices than the field strain eggs. Interestingly, the field strain eggs in 900 MHz and 18 GHz groups displayed greater width dimensions than their laboratory counterparts.

Several conclusions for a comprehensive analysis of the measurements are made based on the results as follows:

1. The control group eggs from laboratory strains acquired an average length of $745.63 \pm 19.82$ μm, a width of $202.42 \pm 16.65$ μm, an area of $751.80 \pm 23.75$ μm$^2$, and an egg index of $3.70 \pm 0.24$ μm.

2. The control group eggs from field strains computed an average length of $672.71 \pm 36.13$ μm, a width of $200.14 \pm 35.26$ μm, an area of $703.20 \pm 62.13$ μm$^2$, and an egg index of $3.47 \pm 0.70$ μm.

3. The 900 MHz exposed eggs from laboratory strains demonstrated an average length of $726.87 \pm 23.34$ μm, a width of $194.32 \pm 19.48$ μm, an area of $726.87 \pm 23.34$ μm$^2$, and an egg index of $3.76 \pm 0.40$ μm.

4. The 900 MHz exposed eggs from field strains exhibited an average length of $702.21 \pm 61.59$ μm, a width of $211.02 \pm 52.44$ μm, an area of $678.35 \pm 38.58$ μm$^2$, and an egg index of $3.56 \pm 1.09$ μm.

5. The 18 GHz exposed eggs from laboratory strains displayed an average length of $726.89 \pm 34.52$ μm, a width of $198.88 \pm 27.49$ μm, an area of $727.92 \pm 36.17$ μm$^2$, and an egg index of $3.71 \pm 0.50$ μm.

6. The 18 GHz exposed eggs from field strains generated an average length of $689.49 \pm 18.98$ μm, a width of $209.29 \pm 26.21$ μm, an area of $685.14 \pm 20.55$ μm$^2$, and an egg index of $3.34 \pm 0.40$ μm.

These detailed measurements offered valuable information about the complex differences in egg morphologies caused by RF exposure in *Ae. aegypti* laboratory and field strains. Table 4 tabulates the morphometric analysis results comparing two distinct RF radiation groups. Consequently, no significant differences were identified for the laboratory and field strains regarding egg length, width, area, and index ($p > 0.05$). These findings implied an insufficient correlation between the unexposed and RF-exposed eggs. No discernible correlation was observed between low-level (900 MHz) and high-level RF exposures (18 GHz).

## The RF exposure effect on the hatching responses of *Ae. aegypti* mosquito eggs

This section provides insights into the experimental results examining the hatching responses of *Ae. aegypti* eggs from laboratory strains following various RF radiation exposures. Fig 2 portrays the hatching responses of *Ae. aegypti* eggs after RF irradiation, which the results highlight that all three radiation doses lead to hatching rates above 50% (see Fig 2A). The control, 900 MHz, and 18 GHz groups demonstrated hatching rate values of $61.67 \pm 29.87\%$, $79 \pm 10.54\%$, and $68.67 \pm 14.74\%$, respectively. Thus, the control group had the lowest hatching rate, while the 900 MHz group presented the highest. Even though the proportion of hatching rates varied across different RF exposures, the data indicated that the statistical analysis did not detect any significant differences between all the groups ($p = 0.87$).

**Table 4. Comparison summary of the egg morphometric analysis based on RF exposures for the *Ae. aegypti* laboratory and field strains.**

| Strain | Parameter | Exposure | *p*-value |
|---|---|---|---|
| **Laboratory strain** | Length (μm) | Control and 900 MHz | 0.12 |
| | | Control and 18 GHz | 0.28 |
| | | 900 MHz and 18 GHz | 0.83 |
| | Width (μm) | Control and 900 MHz | 0.31 |
| | | Control and 18 GHz | 0.73 |
| | | 900 MHz and 18 GHz | 0.68 |
| | Area (μm$^2$) | Control and 900 MHz | 0.10 |
| | | Control and 18 GHz | 0.22 |
| | | 900 MHz and 18 GHz | 0.93 |
| | Egg index | Control and 900 MHz | 0.61 |
| | | Control and 18 GHz | 0.94 |
| | | 900 MHz and 18 GHz | 0.80 |
| **Field strain** | Length (μm) | Control and 900 MHz | 0.21 |
| | | Control and 18 GHz | 0.31 |
| | | 900 MHz and 18 GHz | 0.55 |
| | Width (μm) | Control and 900 MHz | 0.45 |
| | | Control and 18 GHz | 0.38 |
| | | 900 MHz and 18 GHz | 0.88 |
| | Area (μm$^2$) | Control and 900 MHz | 0.29 |
| | | Control and 18 GHz | 0.70 |
| | | 900 MHz and 18 GHz | 0.43 |
| | Egg index | Control and 900 MHz | 0.75 |
| | | Control and 18 GHz | 0.58 |
| | | 900 MHz and 18 GHz | 0.44 |

Fig 2B depicts the hatching periods of *Ae. aegypti* eggs from a laboratory strain under various RF radiation levels. The control, 900 MHz, and 18 GHz groups produced mean hatching time values of 6.5 ± 4.18 days, 6 ± 3.89 days, and 5.5 ± 3.61 days, respectively. Therefore, the 18 GHz group acquired the shortest mean hatching time, while the control group revealed the longest. Despite some variances in the hatching periods, the hatching period analysis across various RF radiation levels did not demonstrate any significant differences between all the groups (*p* = 0.99).

Another finding from this analysis offered valuable information regarding the range of hatching periods associated with RF radiation exposure. The control, 900 MHz, and 18 GHz groups computed minimum and maximum hatching period ranges of less than 24 h to 11 days, less than 24 h to 12 days, and less than 24 h to 11 days, respectively. Therefore, the 18 GHz group demonstrated the fastest hatching times. The boxplot also emphasised that the RF-EMF exposure could reduce hatching durations compared to unexposed eggs (control).

## The RF exposure effect on the developmental stages of *Ae. aegypti* mosquito eggs

Fig 3 displays the developmental stage results of the *Ae. Aegypti* eggs under varying RF radiation levels, ranging from the larval phase to adulthood. A comparison of the larval phase onset between the control, 900 MHz, and 18 GHz groups was conducted (see Fig 3A). The control and 900 MHz groups revealed a similar proportion of eggs hatching into the first larval stage

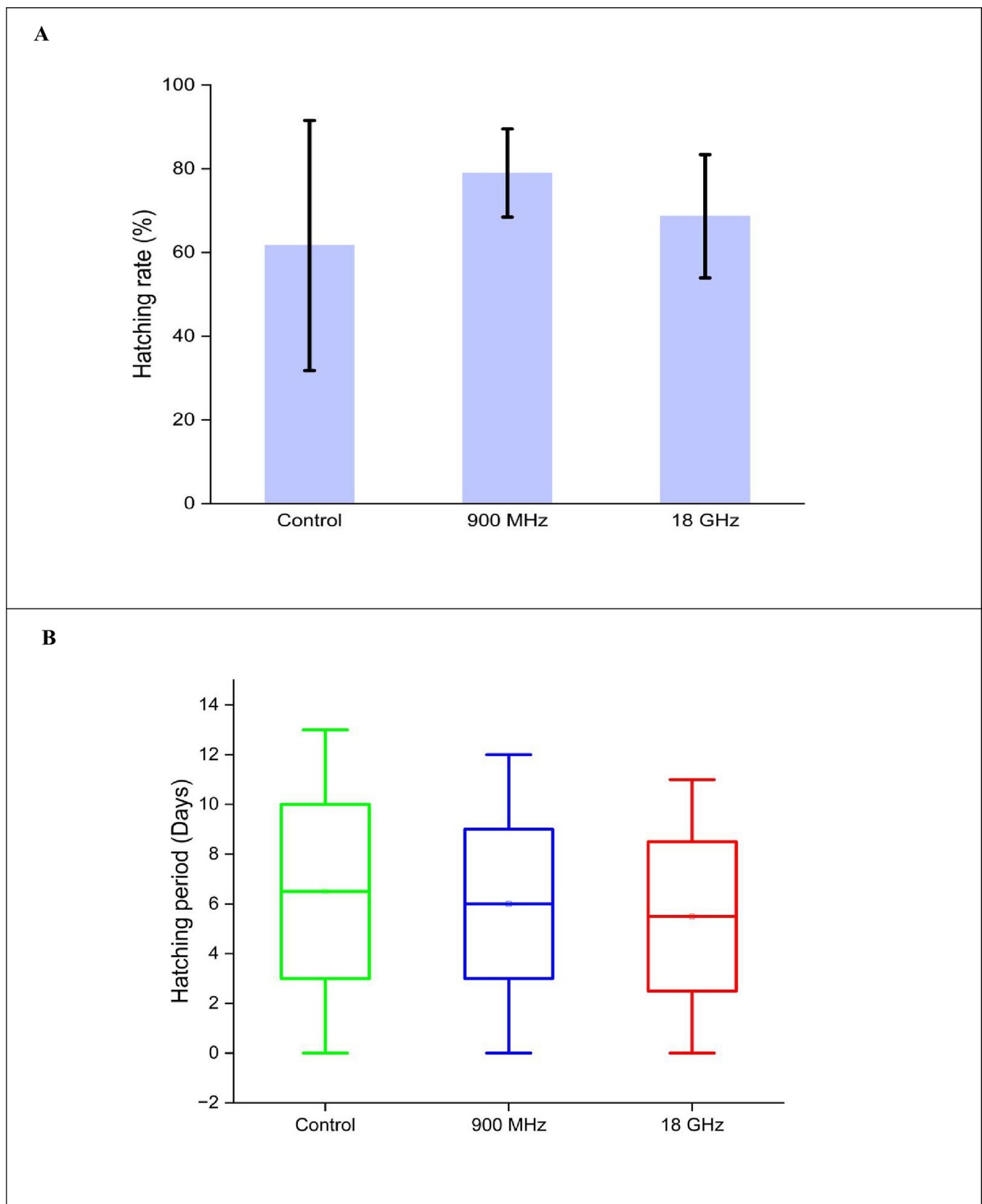

**Fig 2.** The egg-hatching results of the *Ae. Aegypti* eggs involving the (A) hatching rates (%) and (B) hatching periods (day) under various RF radiation levels.

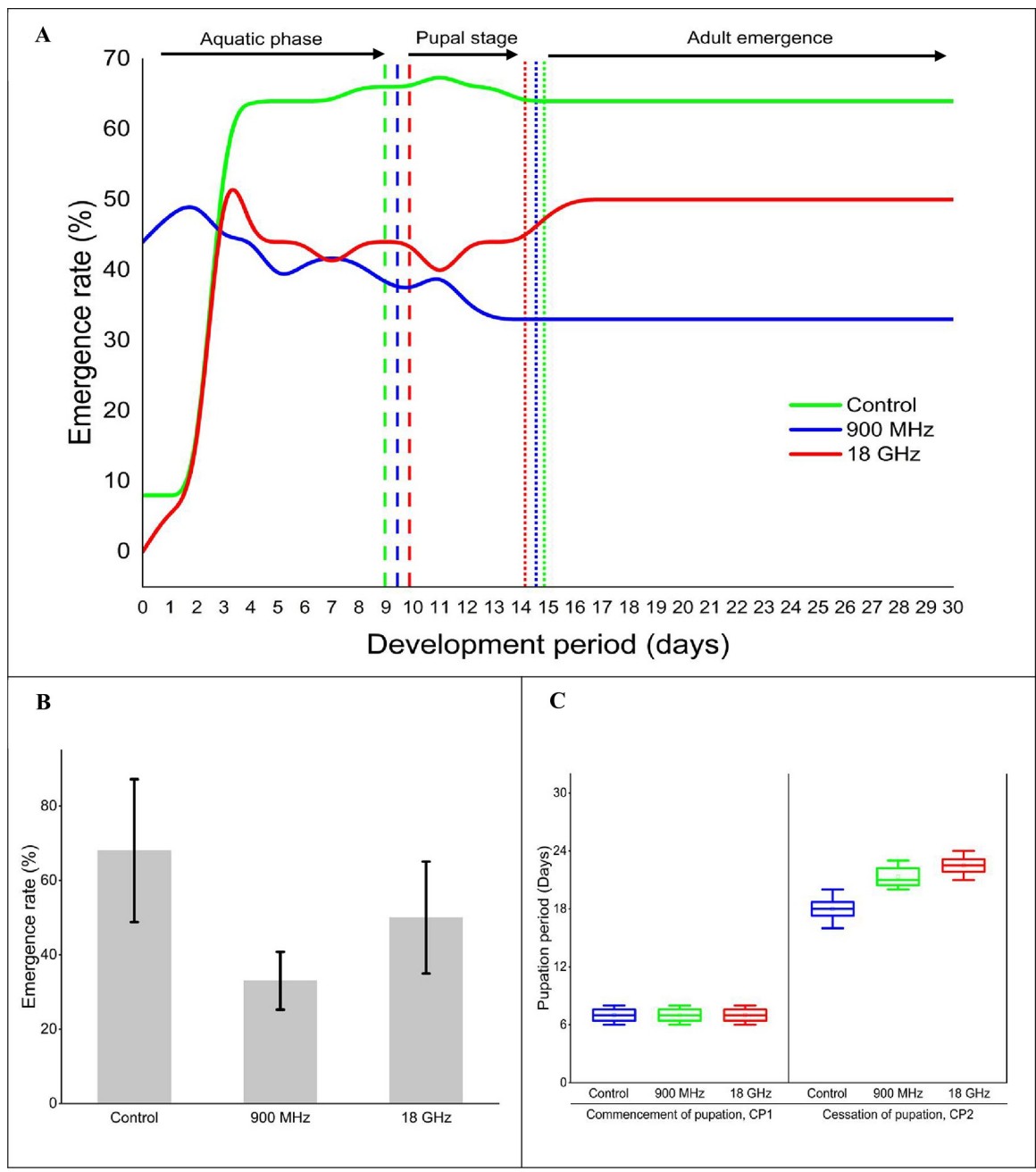

**Fig 3.** The developmental stages of *Ae. aegypti* from larva stage to adulthood involving (A) the development trends of immature exposed under varying RF radiation levels (%), (B) the adult emergence rates (%), and (C) the commencement and cessation periods of pupation (days).

within 24 h at 8% and 44%, respectively. On the contrary, the 18 GHz group experienced an onset on the first day, resulting in 6% of eggs successfully hatching into the first larval stage. The larval development progress to the pupal stage for the control, 900 MHz, and 18 GHz groups were 9 ± 1.73 days, 9.33 ± 0.76 days, and 9.83 ± 0.58 days. This outcome indicated that the control group displayed the shortest period compared to the 900 MHz 18 GHz groups.

Although the larval development period across the various RF exposures varied, the data indicated that the statistical analysis did not detect any significant differences between all the

groups ($p = 0.65$). The time required to complete the emergence of adult mosquitoes for the control, 900 MHz, and 18 GHz groups were 14.78 ± 2.66 days, 14.56 ± 0.97 days, and 14.23 ± 1.04 days. Even though some variances regarding the emergence period were recorded among various RF radiation levels, the statistical analysis from the emergence period analysis suggested that no significant differences were observed between all groups ($p = 0.40$).

Fig 3B portrays the adultisation rates of all the groups. The emergence rates for the control, 900 MHz, and 18 GHz groups were 68 ± 1.33%, 33 ± 2.77%, and 50 ± 9.61%, respectively. This finding implied that the control group possessed the highest adult emergence rate, while the 900 MHz group revealed a lower adult emergence rate. Compared to the control and 18 GHz groups, the data suggested that a 900 MHz RF exposure could produce a more pronounced effect on the growth of *Ae. aegypti* mosquitoes. The variations in adult emergence rates across various RF radiation levels also indicated that the statistical analysis identified a significant distinction between all the groups ($p = 0.03$). Despite the emergence rate was significantly different between the control and 900 MHz groups ($p = 0.02$), no statistically significant difference was observed in the emergence rate between the control and 18 GHz groups ($p = 0.09$) or between the 900 MHz and 18 GHz groups ($p = 0.32$).

Fig 3C displays the pupation periods of all the groups. The pupation ($CP_1$) onset remained consistent across all groups, lasting 7 ± 1 days. This outcome implied that the statistical analysis did not detect significant group variations ($p = 0.99$). Meanwhile, the completion of pupation ($CP_2$) rates for the control, 900 MHz, and 18 GHz groups were 18 ± 1.58 days, 21.33 ± 1.53 days, and 22.5 ± 1.29 days, respectively. Thus, the control group exhibited the shortest pupation period, while the 18 GHz group demonstrated the longest (see boxplot). Nevertheless, the data indicated that the statistical analysis failed to identify significant distinctions between all the groups ($p = 0.08$). The variations in the cessation of pupation suggested that RF exposures could demonstrate a lesser impact on the stages of pupal development.

## Discussion

The current worldwide increase in vector-borne diseases has led to the widespread presence of illnesses such as dengue fever, Zika, yellow fever, and chikungunya. This process has been exacerbated by rapid urbanisation and high population density. The *Ae. aegypti* mosquito is a primary vector of many diseases, particularly well-adapted to urban environments. This study explored the various RF radiation level effects on the life cycle and characteristics of *Ae. aegypti* mosquitoes, focusing on egg morphology and development. Consequently, the *Ae. aegypti* laboratory strains consistently acquired greater mean egg weights than the field strains across all experimental conditions.

The mean egg weight variations were attributed to differing radiation exposure levels, with the field strain eggs producing a modest weight reduction between all the groups. This result indicated that the laboratory strain was more resistant to the impacts of RF radiation. The *Ae. aegypti* laboratory strains presented a greater egg weight than the field strains, raising interesting questions about the mechanisms determining mosquito reproductive features. This observed variation was attributed to environmental influences, genetic adaptation, and artificial selection. Considering that mosquitoes were exposed to stable conditions, abundant food supplies, and regulated temperature with humidity levels, these factors promoted optimal nutrition for female mosquitoes. This nutrition ensured the availability of resources to develop their eggs. Heavier eggs were mainly observed in the laboratory strains due to the favourable environmental conditions [4].

This study suggested a plausible correlation between RF radiation exposure and the mean weight of mosquito eggs. A study by Urbanski *et al.* reported that the differences in egg weight

measurements were associated with the natural process [26]. The study revealed that the freshly laid *Ae. aegypti* eggs gradually absorbed water over the initial 16 h. Similarly, a study by Vargas *et al.* denoted that *Ae. aegypti* eggs were prone to dehydration, which caused variations in their weight and potentially affected their egg viability [27]. Inadequate information regarding the typical egg weight of mosquitoes was also observed. Previous studies primarily concentrated on specific species (such as *Ae. albopictus*), which only focused on desiccation resistance. Therefore, comparing the conclusions of this study directly with those of earlier studies was challenging. Nonetheless, this study represented an essential initial step that laid the groundwork for future studies concerning the mean weight of mosquito eggs and the potential implications of RF radiation exposure on their biological development.

The egg lengths between the *Ae. aegypti* laboratory and field strains demonstrated differences in all the groups. Regardless of RF exposure, the eggs from the laboratory strains consistently acquired higher length measurements than the field strains from all the groups. Conversely, the unexposed field strains produced longer egg lengths than the RF-exposed laboratory counterparts. Meanwhile, the length of the eggs remained consistent between the laboratory strains of the 18 GHz group and the non-irradiated eggs. Despite that this observation closely aligned with a study by Suman *et al.* [24], this observation differed from studies by Mundim-Pombo *et al.* and Bova *et al.* [25, 28]. The differences observed were ascribed to variations in geographical origin, topographical features of the study sites, diverse climatic conditions, and other contributing factors. This study also revealed that the eggs from the laboratory and field strains for all groups highlighted narrower dimensions, which contradicted studies by Suman *et al.*, Mundim-Pombo *et al.*, and Win *et al.* [24, 25, 29]. In contrast, the egg index values in this study closely resembled the findings of Mundim-Pombo *et al.*'s study [25]. Overall, these studies covered a wide range of geographic regions (such as Australia and Brazil), emphasising the possible impact of morphological characteristics of mosquito eggs.

A significant impact of the various RF radiation levels on the mosquito egg-hatching process was presented in this study. The 900 MHz and 18 GHz eggs exhibited the highest hatching rates compared to the control group. Intriguingly, these eggs also hatched relatively quickly. Even though the 900 MHz group demonstrated the highest mean hatching rate, a more moderate hatching period was required. Alternatively, the control group possessed a reasonable hatching rate but needed the most time to finish the hatching process. Thus, the RF exposure levels directly impacted the mean hatching rate and hatching period. Specifically, the hatching process was slower in the 900 MHz and the control group than in the 18 GHz group.

Two studies by Ranathunge *et al.* and Du *et al.* supported the outcome of this study, which noted a substantial decline in hatching rates with increasing X-ray exposure [30, 31]. The studies also examined the effects of pupal irradiation on honey bees' sterility and pupal development from cell phone RF levels of 40 Gy and 60 Gy. When regular male bees were paired with irradiated pupae-based adult female bees, the mating outcomes (egg production and hatching rates) were significantly worse than the control groups. Another study by Odemer *et al.* established a positive correlation between the radiation dose, the quantity of eggs, and hatching rates [32]. Meanwhile, a study by Sahib *et al.* described that GSM technology emitting 900 MHz radiation from mobile phones resulted in reduced bee strength and a lower queen egg laying rate [10]. The study revealed that the test colonies with exposed queens exhibited reduced daily egg production compared to the control colonies. These results were ascribed to the detrimental impact of RF radiation on cells and DNA, which could lead to aberrant development or embryo mortality.

Overall, these findings offered a broad comprehension, and the hatching rates of *Aedes* eggs could vary based on the specifics of the studies and experiments. The observed variation in hatching percentages of mosquito eggs was ascribed to their storage duration. Over time, the

viability of preserved eggs could decline, resulting in a lower hatching success rate. The decrease in viability could be attributed to the eggs collapsing or the embryo perishing as the storage duration increased [28]. Although the hatching rates were observed to be reduced, they remained at or above the control group level. Therefore, the reduced hatching rates were not directly caused by RF exposure but attributed to factors linked to prolonged storage and the associated decline in egg viability.

This study demonstrated that the varying RF radiation levels significantly impacted this species' developmental abilities. Compared to the 900 MHz and 18 GHz groups, the control group exhibited the longest development time from egg hatching to adult emergence. Even though the 900 MHz group presented the fastest development (from egg to larval) of egg hatching during the aquatic phase, the lowest adult emergence rate was observed compared to the control and 18 GHz groups. These findings were consistent with previous studies on *Ae. albopictus* development and the EMF exposure effects.

A study by Atli *et al.* discovered that exposure to microwave EMF resulted in a pupation delay without affecting the rate at which pupation occurred [33]. On the contrary, a study by Ernawan *et al.* observed that while the pupation rate in *Ae. aegypti* mosquitoes decreased as the gamma irradiation dose increased, no significant statistical difference was presented between the control group and the 70 Gy dose. Similarly, a study by Aldridge *et al.* reached a similar conclusion, determining the gamma irradiation dose administered to *Ae. aegypti* pupae did not significantly impact the percentage of successfully hatched adults [22]. Another study by Du *et al.* discovered no significant differences in pupal hatching rates of *Ae. albopictus* compared to the control group (regardless of age groups and X-ray irradiation dosages) [31].

The notable phenomenon identified in this study is that *Ae. aegypti* eggs exposed to RF irradiation displayed a different developmental pattern than non-irradiated eggs. The 900 MHz group particularly computed faster adult development than the 18 GHz group. A study by Ernawan *et al.* proposed that the potential impact of RF radiation on egg physiology and biochemistry could affect the rate of egg growth [34]. The RF radiation could also influence the physiology and biochemistry of adult mosquitoes and their larvae, potentially affecting developmental rates. Nonetheless, further studies should be performed to thoroughly comprehend the full impact of RF radiation on insect development and the underlying mechanisms involved.

## Conclusion

This study successfully comprehended the influence of RF exposure on the development of *Aedes* mosquitoes. The 900 MHz RF exposure accelerated the hatching process of *Ae. aegypti* mosquitoes and increased the percentage of adult emergence. These findings represented an essential initial stage in understanding the impact of RF radiation on *Aedes* mosquito populations, providing vital insights into the population dynamics. Nevertheless, the potential variability results under distinct study conditions involving RF exposure variation types and the mosquito species studied were necessary. Thus, additional investigation was desirable and crucial to understand the consequences of RF exposure comprehensively on *Ae. albopictus* and determine the most efficient approaches for identifying the most effective strategies for dengue vector control. Studies are actively investigating the influence of RF exposure on insects (particularly mosquitoes) due to data suggesting that it can impact hatching and developmental processes. Hence, further studies should be conducted to fully comprehend the scope of these effects and clarify their practical relevance in preventing dengue spread. These ongoing studies are pivotal in learning the potential utility of RF exposure to mitigate the spread of dengue disease and in establishing the most efficacious approaches for translating this knowledge into practical control measures.

## Acknowledgments

The authors would like to express their deep gratitude to the organisations instrumental in completing this project. A special note of appreciation goes to the Faculty of Health Sciences, Universiti Teknologi MARA (UiTM) and Faculty of Medicine and Health Sciences, Universiti Putra Malaysia for their invaluable technical support and guidance during this research project.

## Author Contributions

**Conceptualization:** Nik Muhammad Hanif Nik Abdull Halim, Nazri Che Dom, Nurul Huda Abd Rahman, Rahmat Dapari.

**Data curation:** Nik Muhammad Hanif Nik Abdull Halim, Rahmat Dapari.

**Formal analysis:** Nik Muhammad Hanif Nik Abdull Halim.

**Investigation:** Nik Muhammad Hanif Nik Abdull Halim, Rahmat Dapari.

**Methodology:** Nik Muhammad Hanif Nik Abdull Halim, Alya Farzana Mohd Jamili, Nazri Che Dom, Nurul Huda Abd Rahman, Rahmat Dapari.

**Project administration:** Nik Muhammad Hanif Nik Abdull Halim.

**Supervision:** Rahmat Dapari.

**Validation:** Nik Muhammad Hanif Nik Abdull Halim.

**Visualization:** Nik Muhammad Hanif Nik Abdull Halim.

**Writing – original draft:** Nik Muhammad Hanif Nik Abdull Halim, Alya Farzana Mohd Jamili, Nazri Che Dom.

**Writing – review & editing:** Nik Muhammad Hanif Nik Abdull Halim, Alya Farzana Mohd Jamili, Nazri Che Dom, Nurul Huda Abd Rahman, Zana Jamal Kareem, Rahmat Dapari.

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
