## [Decision Letter · Decision Letter 0]

13 Dec 2023

PONE-D-23-36469Impact of Varying Radiofrequency Exposure on Aedes aegypti (Diptera: Culicidae) DevelopmentPLOS ONE

Dear Dr. Dapari,

Thank you for submitting your manuscript to PLOS ONE. After careful consideration, we feel that it has merit but does not fully meet PLOS ONE’s publication criteria as it currently stands. Therefore, we invite you to submit a revised version of the manuscript that addresses the points raised during the review process.

Dear Dr. Dapari,

Please refer to reviewers comments to improve your manuscript. The study of the effect of radiation on the yellow fever mosquito could be of interest for scientific community. After addressing all reviewers comments, you should emphasize the usefulness and the future of your findings at practical manner.

Regards

We look forward to receiving your revised manuscript.

Kind regards,

Rachid Bouharroud

Academic Editor

PLOS ONE

Journal Requirements:

Additional Editor Comments:

Dear Dr. Dapari,

Please refer to reviewers comments to improve your manuscript. The study of the effect of radiation on the yellow fever mosquito could be of interest for scientific community. After addressing all reviewers comments, you should emphasize the usefulness and the future of your findings at practical manner.

Regards

Reviewers' comments:

Reviewer's Responses to Questions

**Comments to the Author**

1. Is the manuscript technically sound, and do the data support the conclusions?

Reviewer #1: Partly

Reviewer #2: Yes

2. Has the statistical analysis been performed appropriately and rigorously? 

Reviewer #1: Yes

Reviewer #2: No

3. Have the authors made all data underlying the findings in their manuscript fully available?

Reviewer #1: Yes

Reviewer #2: Yes

4. Is the manuscript presented in an intelligible fashion and written in standard English?

Reviewer #1: No

Reviewer #2: Yes

5. Review Comments to the Author

Reviewer #1: General comments:

1. The author should be including more details about nutrients used for larval development, how were they supplied? weight? volume? blood meal with mice, clarify if the expected time was to allow mating, ensure insemination of the females, time for oogenesis.

Clarify, if the replications of the experiment design were at the same time, at different times on the same day, or on different days. How often was the water changed?

How did you ensure that the sample size (unhatched eggs or larvae) was maintained after this manipulation?

2. The authors provide a comprehensive overview of the data analysis.

3. The manuscript presents grammar errors, and mistakes in usage of words, it is recommended to check the grammar throughout the document.

4. It is suggested to make the introduction completer and more concise, "what are the most important Aedes species worldwide? Transmitter dengue and other arboviruses, which ones? What are the traditional and innovative control methods?"

5. Did exposure to 18 GHz show the longest development time until pupation? How do the authors explain this regarding what they conclude?

6. The authors conclude “Our research has revealed that higher doses of RF radiation can expedite the hatching process of Aedes mosquitoes, while concurrently slowing down their overall developmental rate”, what does it refer to? Larval development time? Life cycle? The results indicate larval development times between 9-10 days, and until the adult emergence between 14-15 days. This statement is contradictory.

7. closed laboratory populations (laboratory strain) are more conserved and adapted to controlled and favorable conditions, so their performance is usually higher. The authors should justify a little more the findings regarding the parameters of size and weight of the eggs since they could be directly influenced by the line of mosquitoes, their adaptations, and the diet of the females.

8. Another important consideration about the variability of hatching percentages is that they could be influenced by their storage time, given that as the days go by, they can lose viability, they can collapse, and the embryo can die. Hatching was slower, but it was not less than the control group. The reduced % hatching may not be due to RF, but rather to the aforementioned factors.

9. The authors point out that there is a positive correlation between lower exposure to RF, indicating shorter pupation time and adult emergence; however, it is contradictory when observing a longer pupation time at 38 GHz compared to 900 MHz, but not for emergence, in which control had the highest %, followed by 18GHz and 900 MHz

Reviewer #2: The reviewed article investigates the impact of radiofrequency (RF) electromagnetic fields on Aedes aegypti mosquitoes, prevalent in indoor environments with higher RF exposure risk. The study exposes mosquito eggs to RF radiation at different frequencies and meticulously monitors hatching responses, development times, and other parameters. Results suggest that RF radiation affects the duration of mosquito growth and induces changes in egg morphology. The study emphasizes the sensitivity of A. aegypti egg morphology and development during the aquatic phase to RF radiation, potentially altering their life cycle and, consequently, raising broader ecological concerns.

The paper is informative, generally well written. However, it can be greatly improved by improving statistical analyses. Without deeper tests, it would be very risky to make some conclusions like the existence of an effect of RF on eggs weight or the differences between wild and laboratory strains.

In the summary, you may consider rephrasing the last sentence for clarity:

Original: "Insects closely associated with humans, like mosquitoes, may experience increased RF absorption and dielectric heating."

Suggestion: "Insects closely associated with humans, such as mosquitoes, may experience increased RF absorption and dielectric heating."

Ae. Aegypti � A. aegypti

Keywords: Eliminate RF as it’s difficult to understand what it refers to.

Ae. Albopictus � A. Albopictus

“The use of a completely randomized design ensured that treatments were assigned to experimental units in a random and unbiased manner, enhancing the reliability and validity of the results. This design facilitated a systematic and controlled examination of how exposure duration and population variations impact the Ae. aegypti population, ultimately enhancing the study's comprehensiveness and reliability.’’

This section could be deleted as it doesn’t provide substantial information.

P18 and P19: Are the difference between egg weight of Ae. aegypti statistically significant or?

P22: The interval between the first and last hatching is too pong which makes the assessment of the effect of RF on hatching rate irrelevant. To overcome this situation, it’s recommended to find a way to synchronize the development stage of different batches of eggs. On possibility is to reduce the time between eggs harvests. In addition, the statistical analysis should be more advanced in the text and in the graphs.

P24: It’s not clear if the difference in the weight is due to the strain (wild/laboratory) or to the radiation.

6. PLOS authors have the option to publish the peer review history of their article (what does this mean?). If published, this will include your full peer review and any attached files.

Reviewer #1: No

Reviewer #2: No

---

## [Author Response · Author response to Decision Letter 0]

23 Jan 2024

I would like to submit the revised manuscript entitled “The Impact of Radiofrequency Exposure on Aedes aegypti (Diptera: Culicidae) Development” by Rahmat Dapari to be considered for publication as “an original article” in the PLOS ONE.

Below are the list of changes and rebuttal directed towards each point raised by the reviewer

Reviewer 1

No Reviewer comments Action taken

A1a The author should be including more details about nutrients used for larval development, how were they supplied? weight? volume? blood meal with mice, clarify if the expected time was to allow mating, ensure insemination of the females, time for oogenesis. All stages of larvae were subsequently fed daily

until pupation, with an increasing feeding regime for each stage per day consisting of 0.06 to 0.12 mg per larva in L1-L2, 0.24 mg in L3, and 0.48 mg in L4. After a 72-hour mating period, a blood meal was administered by introducing a confined white mouse into the mosquito cage for 12 hours to ensure that the female mosquitoes were sexually mature and had laid eggs.

A1b Clarify, if the replications of the experiment design were at the same time, at different times on the same day, or on different days. How often was the water changed?

 The replications were at the same time on the same day. The pans were checked daily at the same hour of the day. Debris was removed using plastic pipettes, and water lost through evaporation was replaced daily to maintain constant levels

A1c How did you ensure that the sample size (unhatched eggs or larvae) was maintained after this manipulation?

 Standardized blood feeding (duration), control mating, and ensuring optimal environmental conditions of laboratory

A2 The authors provide a comprehensive overview of the data analysis.

 The comment given by reviewer are taken

A3 The manuscript presents grammar errors, and mistakes in usage of words, it is recommended to check the grammar throughout the document.

 The comment given by reviewer are taken and already corrected as suggestion.

A4 It is suggested to make the introduction completer and more concise, "what are the most important Aedes species worldwide? Transmitter dengue and other arboviruses, which ones? 

What are the traditional and innovative control methods?"

 A. aegypti, commonly known as the 'yellow fever mosquito' [3], plays a pivotal role in transmitting dengue fever while A. albopictus also known ‘Asian tiger mosquito’ is a secondary vector for these diseases.

To effectively manage Aedes mosquito populations traditional methods include source reduction, larvicides and biological control were applied. Recently, innovative approaches involve genetic control using Wolbachia bacteria to hinder virus transmission, employing spatial repellents, and utilizing GIS and remote sensing for larval source management. All these methods for a comprehensive approach to mosquito control.

A5 Did exposure to 18 GHz show the longest development time until pupation? How do the authors explain this regarding what they conclude?

 The boxplot underscores that A. aegypti exposed to 18 GHz radiation did indeed show the longest development time until pupation.

A6 The authors conclude “Our research has revealed that higher doses of RF radiation can expedite the hatching process of Aedes mosquitoes, while concurrently slowing down their overall developmental rate”, what does it refer to? Larval development time? Life cycle? The results indicate larval development times between 9-10 days, and until the adult emergence between 14-15 days. This statement is contradictory.

 The sentence had been changed with this statement. Our research has revealed that 900 MHz RF exposure enhancing the hatching process of A. aegypti mosquitoes, while concurrently expedite the percentage of adult emergence.

A7 Closed laboratory populations (laboratory strain) are more conserved and adapted to controlled and favourable conditions, so their performance is usually higher. The authors should justify a little more the findings regarding the parameters of size and weight of the eggs since they could be directly influenced by the line of mosquitoes, their adaptations, and the diet of the females.

 It was discovered that the laboratory strains of A. aegypti had more egg weight than the field strain, raising intriguing issues about the underlying processes determining mosquito reproductive features. This variation could be explained by a mix of environmental influences, genetic adaptation, and artificial selection. Mosquitoes experience stable conditions, abundant food supplies, and controlled temperature and humidity levels in a controlled laboratory setting, encouraging optimal nutrition of the females and resource availability for egg development. In the laboratory strain, this favourable environment may result in heavier eggs

A8

 Another important consideration about the variability of hatching percentages is that they could be influenced by their storage time, given that as the days go by, they can lose viability, they can collapse, and the embryo can die. Hatching was slower, but it was not less than the control group. The reduced % hatching may not be due to RF, but rather to the aforementioned factors.

 The variability in hatching percentages observed in mosquito eggs could be attributed to the duration of their storage. Over time, the viability of stored eggs may diminish, leading to a decrease in hatching success. This reduction could be a result of the eggs collapsing or the embryo perishing as storage time extends [28]. While hatching rates were observed to be slower, they did not fall below those of the control group. Therefore, it's plausible that the lower hatching percentages are not necessarily a consequence of radiofrequency (RF) exposure, but rather due to the factors related to prolonged storage and the associated decline in egg viability.

A9 The authors point out that there is a positive correlation between lower exposure to RF, indicating shorter pupation time and adult emergence; however, it is contradictory when observing a longer pupation time at 18 GHz compared to 900 MHz, but not for emergence, in which control had the highest %, followed by 18GHz and 900 MHz

 Concerning the completion of pupation (CP2), the control group exhibited the shortest duration at 18 ± 1.58 days, followed by the 900 MHz group at 21.33 ± 1.53 days and the 18 GHz group at 22.5 ± 1.29 days.

Notably, the group exposed to 18 GHz completed the emergence of adult A. aegypti at 14.23 ± 1.04 days, whereas the 900 MHz group and the control group reached this stage at 14.56 ± 0.97 days and 14.78 ± 2.66 days, respectively.

Fig. 3B reveals the adultization rate, the control group had the higher adult emergence rate, at 68 ±1.33%. The 18 GHz exposure group revealed a lower adult emergence rate of 50 ±9.61% when compared to the control group. Furthermore, the adult emergence rate was 33 ±2.77% lowest in the group exposed to 900 MHz RF level

*Note: Any editing or changing in the main document that had been suggested by the first reviewer will be highlighted in blue colour. 

 

Reviewer 2 (B)

No Reviewer comments Action taken

B1. The paper is informative, generally well written. However, it can be greatly improved by improving statistical analyses. Without deeper tests, it would be very risky to make some conclusions like the existence of an effect of RF on eggs weight or the differences between wild and laboratory strains. The comment given by reviewer are taken and already corrected as suggestion.

B2. In the summary, you may consider rephrasing the last sentence for clarity:

Original: "Insects closely associated with humans, like mosquitoes, may experience increased RF absorption and dielectric heating."

Suggestion: "Insects closely associated with humans, such as mosquitoes, may experience increased RF absorption and dielectric heating." The comment given by reviewer are taken and already corrected as suggestion.

B3. Ae. Aegypti � A. aegypti The comment given by reviewer are taken and already corrected as suggestion.

B4. Keywords: Eliminate RF as it’s difficult to understand what it refers to. The comment given by reviewer are taken and already corrected as suggestion.

B5. “The use of a completely randomized design ensured that treatments were assigned to experimental units in a random and unbiased manner, enhancing the reliability and validity of the results. This design facilitated a systematic and controlled examination of how exposure duration and population variations impact the Ae. aegypti population, ultimately enhancing the study's comprehensiveness and reliability.’’

This section could be deleted as it doesn’t provide substantial information. The comment given by reviewer are taken and already corrected as suggestion.

B6. P18 and P19: Are the difference between egg weight of Ae. aegypti statistically significant or? For instance, in the control group, the egg weight of the laboratory strain measured 12.40 µg ± 0.61, whereas the field strain recorded 11.97 ± 1.43 µg and the difference was statistically significant with p = 0.04. Similarly, in the 18 GHz RF exposure group, the weight of egg from the laboratory strain was 12.23 ±1.90 µg while for the field strain was 11.33 ±0.65 µg and the difference were statistically significant with p = 0.02. In contrast, in the groups exposed to 900 MHz RF, the laboratory strain egg with 12.60 ±0.40 µg was heavier than the field strain egg with 11.37 ±0.56 µg, but the difference was not statistically significant with p = 0.15

B7. P22: The interval between the first and last hatching is too pong which makes the assessment of the effect of RF on hatching rate irrelevant. To overcome this situation, it’s recommended to find a way to synchronize the development stage of different batches of eggs. 

On possibility is to reduce the time between eggs harvests. In addition, the statistical analysis should be more advanced in the text and in the graphs. The F2 eggs were used in the experimental procedures after a storage time between 2 and 3 weeks. The age of egg s batches used in this experiment were suspected to influence the hatching response. Because it was not possible to use eggs of uniform ae throughout the experiment, the age was included as an exclusion criterion

The F2 generation eggs of the A. aegypti strain were randomly allocated to the experimental group to ensure that the age of the eggs would not influence the outcomes

B8 P24: It’s not clear if the difference in the weight is due to the strain (wild/laboratory) or to the radiation. Specifically, when comparing the varied exposures for the laboratory and field strains, no statistically significant difference in egg length was identified between the control group and the groups exposed to 900 MHz or 18 GHz RF (p > 0.05). These findings strongly suggest that the egg weight for both laboratory and field strains were not affected by RF exposure.

*Note: Any editing or changing in the main document that had been suggested by the second reviewer will be highlighted in orange colour. 

Sincerely, 

Dr. Rahmat Dapari 

Principal Investigator,

---

## [Editor Report · Decision Letter 1]

30 Jan 2024

The Impact of Radiofrequency Exposure on Aedes aegypti (Diptera: Culicidae) Development

PONE-D-23-36469R1

Dear Dr. Dapari,

We’re pleased to inform you that your manuscript has been judged scientifically suitable for publication and will be formally accepted for publication once it meets all outstanding technical requirements.

Kind regards,

Rachid Bouharroud

Academic Editor

PLOS ONE
---

## [Editor Report · Acceptance letter]

17 Feb 2024

PONE-D-23-36469R1 

PLOS ONE

Dear Dr. Dapari, 

I'm pleased to inform you that your manuscript has been deemed suitable for publication in PLOS ONE. Congratulations! Your manuscript is now being handed over to our production team.

Kind regards, 

on behalf of

Dr. Rachid Bouharroud 

Academic Editor

PLOS ONE